# Risk Assessment of Spread of the Influenza A Virus in Cows in South Bulgaria

**DOI:** 10.3390/v17020246

**Published:** 2025-02-11

**Authors:** Gabriela Goujgoulova, Koycho Koev

**Affiliations:** 1Risk Assessment Center on Food Chain, 1000 Sofia, Bulgaria; 2Department of Veterinary Microbiology, Infectious and Parasitic Diseases, Faculty of Veterinary Medicine, Trakia University, 6000 Stara Zagora, Bulgaria; koycho.koev@trakia-uni.bg

**Keywords:** HPAI H5N1, dairy cattle, humans, Bulgaria

## Abstract

In this article, we present an assessment of the risk of the potential introduction and spread of highly pathogenic avian influenza (HPAI) in cows in Bulgaria. In the spring of 2024, we witnessed an unprecedented spread of the virus in dairy herds in the USA. This crossing of interspecies barriers by the virus creates a real danger of pandemic manifestations in humans. The continued spread of H5N1 clade 2.3.4.4b in dairy populations and other mammalian species and efficient animal-to-animal transmission increases the risk of infection and subsequent spread of the virus in human populations. According to registers, as of 1 November 2022, a total of 559,544 cattle were bred in Bulgaria. The total number of dairy cows decreased by 5.2% year-on-year to 197,996. Farms breeding dairy cows as of 1 November 2022 were 12,439, which is 22.1% less than the previous year. The production of cow’s milk in 2022 amounted to 748,278 thousand liters. Traditionally, the largest share in the total yield of cow’s milk is occupied by the south-central region with 25.9%, followed by the southeastern region with 18.5%. Due to potential risk factors such as the high concentration of dairy cows in high-risk areas for avian influenza A, the possibility of HPAI jumping the interspecies barrier and spreading in dairy herds in Bulgaria is very high. We therefore set out to assess the risk of virus penetration in these herds.

## 1. Introduction

In recent years, the majority of European countries and a large number of countries around the world have been affected by a pandemic wave of highly pathogenic avian influenza (HPAI), the current one being the most severe in the recent history of the Old Continent.

The virus has an extremely complex molecular taxonomy, phylogenetics, plasticity, mutability, and variability (with evolutionary changes in the molecular structure and integrity of the genome, reassortments, and recombinations between different serovariants of the virus). This is one of the viruses with the most complex behavior ever documented. The adaptation of influenza viruses to a specific biological species does not exclude the possibility that they can adapt and infect other species, creating an ever wider range of hosts beyond the usual ones. In this sense, great danger comes from jumping the species barrier and the possibility of pandemics, including in humans.

For the first time, bovine influenza was reported in 1949 in Japan [1]. Since then, over the years, there have been isolated cases of influenza virus in cattle in Europe and Russia [2,3], but there is an association with the disease and the reduction in milk yield [2,4]. A low seroprevalence of influenza C was reported around 1970. Influenza C virus was isolated from the respiratory tract of cows in 2016. Then, influenza D was also documented in them [1].

On 20 March 2024, the Minnesota Board of Animal Health reported that a goatling raised on a Stevens County farm in west-central Minnesota tested positive for highly pathogenic avian influenza (HPAI). An outbreak of HPAI in poultry had been recently detected on this farm. This was the first case of the virus in a domestic ruminant in the United States. The goats were tested after the farmer noticed unusual deaths in newborn animals, which were found following the culling of poultry as a part of outbreak control in February. Goats and poultry shared a common space as well as the same water source. Ten animals aged 5 to 9 days died [5]. One of the goats was tested at the Minnesota Veterinary Diagnostic Laboratory, where HPAI was confirmed. Subsequent tests at the United States Department of Agriculture (USDA) National Veterinary Laboratory in Ames, Iowa, identified the virus as H5N1 clade 2.3.4.4b, the same virus circulating in wild and domestic poultry in numerous countries, including the United States. Genomic sequencing revealed that samples from the farm’s goats and poultry were genetically close [6].

On 25 March 2024, HPAI was declared in dairy cows in three states—Texas, Kansas, and New Mexico. The infection in cattle led to a decrease in the quality indicators of milk as well as sporadic cases, mainly in older animals. Dead birds were also found at some of the farms, which could explain the source of the virus in unpasteurized milk from sick cattle collected from two dairy farms in Kansas and from one in Texas as well as a throat swab from a cow at another dairy farm in Texas. HPAI viruses from genetic clade 2.3.4.4b were isolated. A few days later, on 29 March 2024, the USDA National Veterinary Laboratory (NVSL) confirmed HPAI in cattle from a farm in Michigan that had recently imported animals from Texas [7,8].

Ly et al., like other researchers, noted that HPAI A(H5N1) viruses of genetic clade 2.3.4.4b found so far among birds worldwide appear very similar to the virus strain of the same genetic clade 2.3.4.4b, originally identified in dairy cattle in Texas and Kansas, where they were first reported on 25 March 2024 [9]. Control and Prevention (CDC) confirmed that one person had tested positive for HPAI. He had been in contact with dairy cattle in Texas that were found to be infected with HPAI (H5N1). The patient reported severe eye redness as the only symptom. This was the second person reported to have tested positive for influenza A (H5N1) viruses in the United States. The previous human case was in 2022 in Colorado. Human infections with avian influenza A virus, including H5N1 viruses, are uncommon but occur sporadically worldwide. The disease can proceed with mild and transient symptoms (conjunctivitis and upper respiratory symptoms) or with severe symptoms such as pneumonia, which can lead to death [10].

By the end of April 2024, the USDA confirmed HPAI H5N1 clade 2.3.4.4b virus on 33 dairy farms in eight states (Kansas, Idaho, Michigan, New Mexico, North Carolina, Ohio, South Dakota, and Texas). Based on specific phylogenetic studies, the USDA confirmed that eight poultry farms in five states (Kansas, Michigan, Minnesota, New Mexico, and Texas) were also infected with the same genotype of HPAI H5N1virus found in dairy cattle [11].

According to the USDA’s Animal and Plant Health Inspection Service, as of early July this year, six states were affected, with the most avian influenza cases reported in Colorado and Iowa, with 23 and 11, respectively. Confirmed cases across the United States numbered 140, affecting 12 states, with 48 cases in the past 30 days at the time of writing. In the United States, more than 130 dairy herds in 12 states tested positive for influenza A as well as 4 alpacas, more than 30 cats, 66 house mice, 1 bobcat, 3 raccoons, a red fox, 1 striped skunk, and 2 Virginia opossums [12].

Positive brain and tissue samples from the five goats for H5N1 from the Minnesota case are cause for concern. Although the virus has not yet been detected in commercial milk, it remains under surveillance due to its potential impact on animal and human health. The current risk to the general public from avian influenza viruses is considered low. People whose work involves exposure to infected birds or animals, including cows, are at greater risk of infection with HPAI [13]

Poultry farming is one of the traditional livestock industries for Bulgaria. The country share of the market for fattened duck liver in Europe is over 20%, and it is known that a large number of low-pathogenic influenza A viruses circulates among mule ducks, which depends upon the way these birds are reared and the biosecurity measures in the farms where they are raised; there is a serious danger of the transmission of viruses by wild birds and the development of epizootics of HPAI. In almost all regions of Bulgaria, there are industrial poultry facilities for laying hens, and there are three regions (Plovdiv, Haskovo, and Stara Zagora) with a high density of duck farms. These facts as well as the passage of large populations of wild birds through the territory of Bulgaria by the main migration routes create potential conditions for the emergence of outbreaks of influenza in birds. From January 2024 until now, 17 outbreaks were announced in Bulgaria in six regions of the country (Veliko Tarnovo, Dobrich, Pazardzhik, Plovdiv, Kardjali, and Haskovo), and there was one case of HPAI in wild birds in the region Haskovo. Peking ducks, mule ducks, broilers, and laying hens are reared in the affected livestock facilities where the presence of HPAI has been confirmed. During this period, two low-pathogenic viruses in ducks were also proven—H6N2 and H1N5.

There are no data on the infection with HPAI of mammalian and human species in Bulgaria, but the risk to mammals should not be ignored [14,15]. Yogurt is a popular and traditional product in the country, and if dairy herds are reduced, this could pose a risk to human health. Bulgarian homemade yogurt is known for its long history of production. Today, there are strict requirements, specialized equipment, and “pure cultures” that exclude any additional microflora naturally found in homemade yogurt. However, many people still make it at home using visual measurements. Due to the lack of studies, whether milk from influenza-infected cows is still infected after being processed at home for yogurt production remains uncertain.

In our paper, we assess the risk of HPAI spread in cows in Bulgaria based on a comparison of isolates from birds in Bulgaria from previous years and isolates from goats, cows, and humans in the current outbreaks in the USA.

## 2. Materials and Methods

### 2.1. The Number and Data of Dairy Farms in Bulgaria

For our study, we used data from agro-statistics of the Ministry of Agriculture and the Register of Group I dairy farms for 2022. The total number of cattle in the country was 559,544, according to registers as of 1 November 2022, which is 5.1% less compared to 2021. The total number of cows decreased by 5.2% on an annual basis, and more a significant decrease was registered in dairy cows, i.e., up to 197,996., and less in beef cows, i.e., up to 163,480. The share of beef cows in the total number of cows is increasing as a result of the ongoing process of specialization of production in cattle breeding, which is characterized by a shift from dairy to meat production. Farms breeding dairy cows as of 1 November 2022 numbered 12,439, 22.1% less than the previous year, with a decrease reported for all categories of farms. In 2022, nearly 39% of all dairy cows in the country were raised on large farms with 100 or more animals.

### 2.2. Sequences of Viruses

The used sequences can be found in the global database for influenza gene sequences (https://gisaid.org/), accessed on 16 March 2011 [16]. We analyzed viruses from Bulgaria from the period 2021–2024, using sequences of HA, NA, and PB2 genes, and compared them with those from mammals from the USA.

### 2.3. FluServer Mutation Tool

For comparison of the mutation, we used the FluServer tool, availabale online: https://gisaid.org/database-features/flusurver-mutations-app/, accessed on 16 March 2011 [16]. With this tool, we screened our influenza sequences for mutations related to host receptor specificity and compared them with sequences from mammals from the USA.

### 2.4. Evolutionary Analysis by Maximum Likelihood Method

For the evolutionary history, we used the maximum likelihood method and the Tamura–Nei model. The trees shown are those with the highest logging probability: HA-6749.91; NA-6106.26; PB2-7819.50. The trees were obtained automatically by applying neighbor-joining and BioNJ algorithms to a matrix of pairwise distances estimated using the Tamura–Nei model. This analysis included 42 for HA and 41 for NA and 42 for PB2 nucleotide sequences. There were a total of 1784 positions in the final dataset for HA, 1467 for NA, and 2316 for PB2. Evolutionary analyses were performed in MEGA11 [17,18].

### 2.5. Philogenic Trees

The graphical view of phylogenic trees was designed by FigTree v1.4.4. software [19].

## 3. Results

### 3.1. Analysis of Data for Dairy Farms Category 1

Cattle breeding is concentrated in Southern Bulgaria, where, in 2022, 67% of the total number of cattle in the country were raised. In the south-central region, more than one-third of the cows are bred, of which 28.2% are dairy, and 41.3% are meat.

Figure 1 and Figure 2 show the percentage distribution of category 1 milk farms.

Figure 1 clearly shows that the concentration of avian influenza outbreaks was in the Plovdiv and Haskovo regions. From the diagram in Figure 2, it can be seen that the most dairy cows are bred in the region of Plovdiv, followed by the regions of Stara Zagora and Haskovo.

### 3.2. Analysis of Viruses Isolated from Birds in Bulgaria and Mammals in the USA in 2024

Since January 2024, 19 outbreaks have been reported in seven regions of Bulgaria (Veliko Tarnovo, Dobrich, Pazardzhik, Plovdiv, Kardzhali, Haskovo, and Yambol), and there were two cases of HPAI in wild birds in the Haskovo region. The affected farms were those with Peking ducks, Muller ducks, broilers, and laying hens. At the end of September, the virus was confirmed at a facility in the village of Stryama, where 6842 pheasants, 375 wild American turkeys, and 3750 partridges were being raised. During this period, two low-pathogenic viruses were also confirmed in ducks—H6N2 and H1N5.

We used MEGA11 to obtain three phylogenetic trees, respectively, for HA, NA, and PB2. For their visualization, we used FigTree v1.4.4. software, and they are presented in the Figure 3, Figure 4 and Figure 5.

Before the appearance of HPAI in dairy cows in Texas, wild birds had been infected, and they are believed to have been the source of the infection in cattle. During the same period, HPAI was also demonstrated in peregrine falcons and skunks in California and New Mexico, respectively.

The HA phylogenetic tree shows a relationship between viruses from alpaca, cows, and common grackles (*Quiscalus quiscula*) and blackbirds (*Turdus merula*) from Texas. This indicates that the viruses share a closer evolutionary relationship than those from other species. The other mammalian isolates are grouped with those from wild birds, suggesting genetic similarity and interspecies transmission (Figure 3).

The HA, NA, and PB2 phylogenetic trees show that the wild bird viruses are closely related to those found in mammals in the USA. The H5N1 avian isolates from Bulgaria are grouped separately, and the LPAIs are completely different because they belong to a different genetic lineage (Figure 3, Figure 4 and Figure 5).

We used FluSever in the GISAID platform and compared used sequences with reference strain A/Anhui/01/2005 (H5N1). In the analysis of viruses isolated from cows, goats, and humans in the USA, the presence of molecular marker T156A was observed in hemagglutinin. In cows, K497R was found in the acid polymerase (PA) and K389R in the base polymerase (PB2). In the human virus from Texas (A/Texas/37/2024), in PB2, there was the molecular marker E627K.

### 3.3. Assessment of the Risk of Introduction and Spread of Avian Influenza A Virus in Dairy Cow Herds

Due to potential risk factors, such as the high concentration of dairy cows in high-risk areas for avian influenza A, the probability of HPAI jumping the interspecies barrier and spreading in dairy herds in Bulgaria is rated on the low (L) to medium (M) level. The risk tends to increase during the autumn period due to the increased migration of wild birds.

The Food Chain Risk Assessment Center assigns the probability of occurrence and spread of HPAI H5N1 on a 6-point scale of risk assessment levels, as follows in Table 1.

## 4. Discussion

In our previous studies, by comparing the HPAI outbreaks, wild birds’ migration maps, and the maps of poultry density, we identified three areas as high risk for avian influenza A disease. They are located in the central part of the country and cover the regions of Plovdiv, Stara Zagora, and Haskovo [14,15]. According to the dairy farm registers, these three regions have the largest number of cattle. More than one-third of the country’s cows are bred in the south-central region, of which 28.2% are dairy and 41.3% are meat. This means that the regions of Plovdiv, Stara Zagora, and Haskovo are high-risk areas not only for poultry but also for dairy cattle.

Influenza A (H5N1) viruses in dairy cows belong to clade 2.3.4.4b genotype B3.13. This genotype is relatively rare in the U.S. but has previously been demonstrated in wild birds and mammals. B3.13 contains the PA, HA, NA, and M gene segments of HPAI (H5N1) from isolates from Europe and the PB2, PB1, NP, and NS gene segments of LPAI viruses from wild birds from the Americas. Genotype B3.13 differs from the virus isolated in March 2024 in goats. The goat outbreak is unrelated to the current cattle outbreak (https://virological.org/t/preliminary-report-on-genomic-epidemiology-of-the-2024-h5n1-Influenza-a-virus-outbreak-in-u-s-cattle-part-1-of-2/970, accessed on 16 March 2011). The results of the phylodynamic analysis indicate that the spread in cattle was probably carried by wild birds.

Genome sequencing of viruses isolated from cattle shows that they have no genetic change that would make them easier to transfer to or between humans, and the CDC considers the risk to humans to remain low. However, there is continued transmission of the virus from cattle to cattle, from cattle to poultry and domestic animal species (domestic cats), and back to wild birds [20]. In May 2024, the same virus circulating in cows caused the infection in an alpaca and on a poultry farm, suggesting mammal-to-bird-to-mammal transmission.

Numerous mutations associated with increased virulence or mammalian adaptation have been identified in the genome. The presence of molecular marker T156A was observed in hemagglutinin. It is a prerequisite for an increase in the ability of the virus to bind to α2,6 sialic acid receptors and an increased possibility of transmission in guinea pigs. Substitution PB2-M631L [21] is found in 99% of the sequences in dairy cows in contrast to birds, where it is rarely observed. Worobey et al. (2024) considered that the M631L mutation in PB2 leads to easier interaction with the host protein ANP32, as does the PB2 E627K mutation in humans [22]. Mutation of PB2 residue 627 from E to K facilitates polymerase activity in mammalian cells. The host transcriptional regulator ANP32A is responsible for this adaptation. In bovine viruses, the presence of PA-K497R and PB2-K389R are indicative of increased polymerase activity in mammalian cells. Molecular marker PB2-E627K in the human virus in PB2 results in increased polymerase activity and increased virulence in mice, contributes to airborne transmission of influenza virus in ferrets and contact transmission in guinea pigs, and causes reduced polymerase activity and replication in avian cell lines and reduced virulence in chickens [23]. Furthermore, the NP gene, which distinguishes B3.13 from all other North American genotypes, may have resulted in a phenotypic change and favored adaptation and spread in this species [22]. Viruses isolated from birds in Bulgaria in 2022–2023 also have T156A in HA and K389R in PB2. Their presence does not pose a direct risk to people’s health, but tracking the dynamics of the virus in the United States is crucial to increase the monitoring of this disease.

It should be noted that people with close or prolonged, unprotected exposures to infected birds or other animals (including large ruminants) or environments contaminated by infected birds or other animals are at a relatively higher risk of infection. The CDC has interim recommendations for prevention, monitoring, and public health research of HPAI (H5N1) viruses.

A team of authors from Finland issued similar recommendations after the July 2023 epidemic caused by the same clade 2.3.4.4b genotype BB in southern and central Finland in foxes, American mink, and raccoon dogs, confirmed in 20 farms. The authors emphasize that although no human cases were detected in this epidemic, there are serious concerns for the future worldwide regarding cross-species, mammal–human transmission [24]. Parums (2024), in his article, also summarized the probability of virus transmission from birds to cattle and vice versa, to other mammals and humans, and, respectively, the possible mutations of the influenza virus with the possibility of adaptation to humans [25].

The antibody response of cattle to influenza virus infection has not been well studied in the past due to the long-standing concept that cattle are not susceptible hosts to influenza A virus. Several studies on bovine influenza D viruses have shown that persistent infections often occur in cattle herds, and existing immunity is short-lived [26]. In this sense, in their comparative analysis in 2024, Gao et al. noted the need to investigate the durability and efficacy of the antibody response against HPAI H5N1 in dairy cows and the possible implementation of vaccination strategies, which is crucial for the control and prevention of future influenza epidemics in cattle [27]. According to the studies of Giménez-Lirola et al. (2024), the virus is present in the milk in the first three weeks after the onset of symptoms, but after that, it is unlikely that the virus will be detected there [28].

Epidemiological studies in dairy farms demonstrate the spread of the virus between herds as well as its return to nearby poultry farms, which is an important interspecies marker for the spread of the virus in different animal populations. The crossing of the interspecies barrier was further demonstrated in April in the state of Texas, when highly pathogenic avian influenza A (H5N1) was detected in several cats living on farms that had HPAI infections in dairy cows, suggesting that the virus had spread to the cats from the affected cows from the raw milk or from the wild birds associated with these farms. Dairy cattle can shed the virus in their milk and therefore can potentially transmit infection to other mammals through unpasteurized milk [29,30].

HPAI H5N1 remains a significant global challenge due to its widespread distribution and high mortality rates. Documented cases of HPAI H5N1 infection in humans, together with recent outbreaks of HPAI H5N1 in various animal species, highlight objective concerns regarding the transmission and spread of this virus among birds, mammals, and humans. This can be realized by small genomic changes (drift) or by the exchange of genomic segments from other influenza A viruses (shift). In this aspect, as Liang (2023) noted that some avian viruses such as H5 and H7 have the potential to affect many people and would have the potential for efficient human-to-human transmission in a potential pandemic scenario [31].

The genetic variation observed among HPAI H5N1 isolates further emphasizes the need for vigilance and continued research to understand virus etiology, origin, distribution, genome structure, phylogeny, evolution, transmission routes, immune response, pathogenesis, and diagnostic and preventive strategies. In this regard, there is a need to compare and publish both serological and sequencing data internationally. Bulgaria, as a country where the HPAI epidemic has persisted for more than three years, is exposed to a huge risk of the virus spreading among mammals.

A limitation of the present study is the lack of data from mammals due to the fact that they were not tested. A new disease monitoring program is due to be introduced next year, which will expand monitoring and will include wildlife sampling.

## 5. Conclusions

There is currently no evidence of infection or even past infection in the cattle population in Europe. To date, circulation of the H5N1 virus in dairy cows has only been reported in the United States. Based on the available information, the World Health Organization (WHO) considers the risk posed by the H5N1 virus to the general human population to be low, and for people who may be exposed to infected animals, such as farmers, veterinarians, and workers in industry, the risk is considered to be low to moderate. The raising of awareness among the cattle sector and the control and monitoring of the disease, however, continues to increase with the aim of reducing and limiting the spread of influenza.

Although the risk for dairy herds in Bulgaria is low, it would be useful to carry out surveys and screenings of farms in high-risk areas of Bulgaria, where previous outbreaks of avian influenza have been reported in poultry and wild birds, in order to prevent a possible outbreak among ruminant herds.

## Figures and Tables

**Figure 1 viruses-17-00246-f001:**
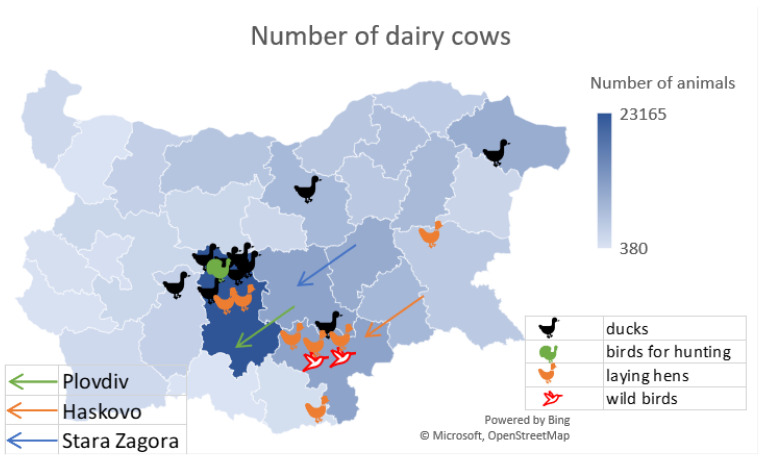
Register of farms for the production of cow’s milk in the category 1 group (https://agri.bg/documents/registar-na-fermite-za-proizvodstvo-na-krave-mlyako-ot-i-grupa accessed on 25 June 2014). Influenza outbreaks for 2024 by bird species have been added to the map.

**Figure 2 viruses-17-00246-f002:**
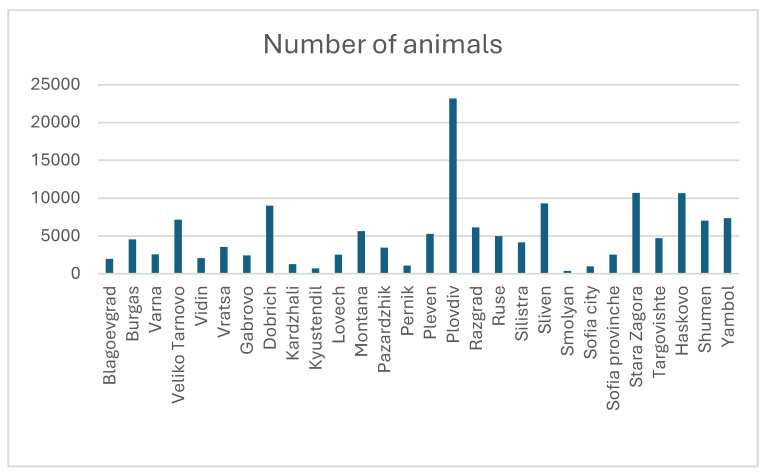
Register of farms for the production of cow’s milk in the category 1 group (https://agri.bg/documents/registar-na-fermite-za-proizvodstvo-na-krave-mlyako-ot-i-grupa, accessed on 16 March 2011).

**Figure 3 viruses-17-00246-f003:**
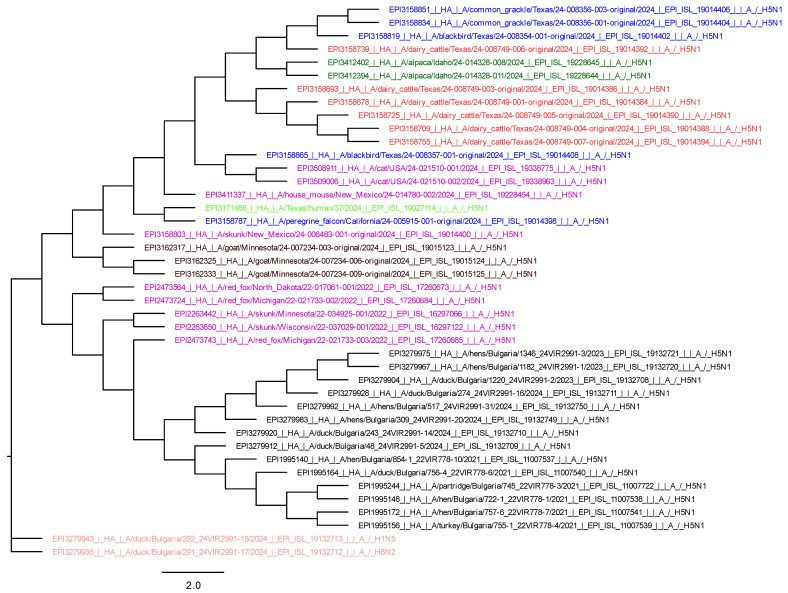
Phylogenetic tree of HA (blue—wild birds in USA; red—dairy cattle in Texas; brown—goat in Minnesota; green—human in Texas; dark green—alpaca; purple—other mammals; black—HPAI isolates from birds from Bulgaria 2021–2024; pink—LPAI from Bulgaria).

**Figure 4 viruses-17-00246-f004:**
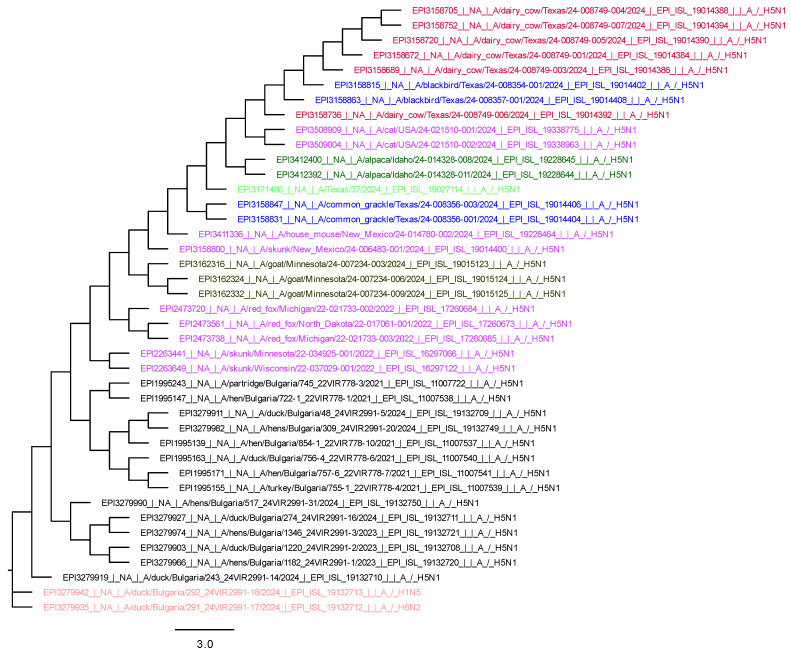
Phylogenetic tree of NA (blue—wild birds in USA; red—dairy cattle in Texas; brown—goat in Minnesota; green—human in Texas; dark green—alpaca; purple—other mammals; black—HPAI isolates from birds from Bulgaria 2021–2024; pink—LPAI from Bulgaria).

**Figure 5 viruses-17-00246-f005:**
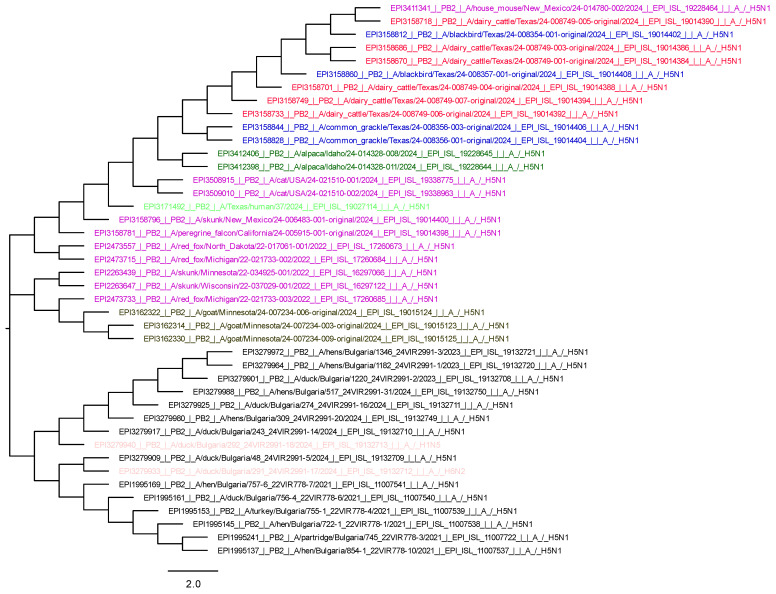
Phylogenetic tree of PB2 (blue—wild birds in USA; red—dairy cattle in Texas; brown—goat in Minnesota; green—human in Texas; dark green—alpaca; purple—other mammals; black—HPAI isolates from birds from Bulgaria 2021–2024; pink—LPAI from Bulgaria).

**Table 1 viruses-17-00246-t001:** Scale of risk assessment levels on a six-point rating scale.

Risk Level	Additional Information
Negligible (N)	Extremely rare; not worth considering
Very Low (VL)	Very low but cannot be excluded
Low (L)	Rare but can occur
Medium (M)	Occurs regularly
High (H)	Occurs frequently
Very High (VH)	Events happen very often

## Data Availability

All sequences are publicly available in GenBank.

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
