# Peer review of "Risk Assessment of Spread of the Influenza A Virus in Cows in South Bulgaria"

_viruses, 2025, doi:10.3390/v17020246_

Round 1

Reviewer 1 Report

Comments and Suggestions for Authors

Dear authors,

Thank you for the manuscript. The idea was to perform risk assessment, however, the results presented in sections 3.2 and 3.3 are very pooorly described and contain only trees and a table. There is no direct risk assessment. Add more detail to this sections and provide risk assessment. Otherwise it looks more like a review paper. Also, please check with care spelling - many misused capital letters. Also improve the quality of figures with phylogenetic trees.

Author Response

Dear Reviewer 1,

Thank you very much for your accuracy and detailed review. It was very helpful to us. I really appreciate the work you have put in. We tried to improve the article and remove most of the remarks. We reconstructed the phylogenetic trees and changed the view for greater distinctness.

To your question: Increasing the ability of the virus to bind to α2,6 sialic acid receptors is not enough to facilitate human-to-human transmission.

Reviewer 2 Report

Comments and Suggestions for Authors

Find please attached.

Author Response

Dear reviewer 2,

Thank you very much for your accuracy and detailed review. It was very helpful to us. We tried to improve the article and remove most of the remarks.

About next reference, they are on the cite of our institution Center of Risk Assesment on Food Chain. The cite is https://corhv.government.bg/. The cite is under reconstruction and maybe it is not working properly.

Goujgoulova G. Scientific opinion on identification risk assessment on high risk areas and vulnerable populations in the poultry sector in Bulgaria regarding highly pathogenic influenza and the birds. CORHV. 2024. https://corhv.government.bg/

Round 2

Reviewer 1 Report

Comments and Suggestions for Authors

Dear authors, thank you for the revised version. I think all the issues are addressed. The only thing to improve is the quality of the phylogenetic trees, currently they are blurry

Author Response

Dear reviewer,

The figures are clearer when they are in pdf. I have added a supplementary file.

Reviewer 2 Report

Comments and Suggestions for Authors

Dear Authors,

This variant of the Article is better than previous. The Figure 1 is improved, but it would be perfectly if you indicate by arrows Plovdiv, Haskovo, and Stara Zagora regions in the Figure 1.

I have also some notes.

1. References

Lines 40-41 contain text “…cases of influenza virus in cattle in Europe and Russia [4], but there is an association with the disease and the reduction in milk yield [7]…” where citing is incorrect.

Both references [4, 7] describe cases in the United Kingdom. However, the introduction of the reference [7] contains information about outbreaks of respiratory disease in cattle caused by influenza virus in some countries that time. The Russia was mentioned among them. Therefore, it would be correct to omit the first square brackets [4] and to give both references in the second one. Finally, it will be so: “…cases of influenza virus in cattle in Europe and Russia, but there is an association with the disease and the reduction in milk yield [4, 7]…”

If you consider mentioning the Russia particularly, then you may add an appropriate reference in the first square brackets [4, X]. The reference X is below.

            References

4. Brown IH, Crawshaw TR, Harris PA, Alexander DJ. Detection of antibodies to influenza A virus in cattle in association with respiratory disease and reduced milk yield. Vet Rec. 1998;143(23):637-638.

7. Crawshaw TR, Brown IH, Essen SC, Young SC. Significant rising antibody titres to influenza A are associated with an acute reduction in milk yield in cattle. Vet J. 2008;178(1):98-102.

X. Fatkhuddinova MF, Kir'ianova AI, Isachenko VA, Zakstel'skaia LIa. [Isolation and identification of the A-Hong Kong (H3N2) virus in respiratory diseases of cattle]. Vopr Virusol. 1973;18(4):474-478. Russian.

2. It seems you have omitted numbering of the article (Parums, D. V., 2024. Concerns as Highly Pathogenic…) between numbers 20 and 21 in line 387. Therefore, your reference numbers in the main text do not match the reference list.

Line 39, 43. Check, please, reference [24]. It is incorrect here.

Check also references [27] in line 56; [23] in line 96; [25, 26] in line 155; [22] in line 157; [30] in line 260; [29] in lines 263 and 272.

            Other notes

Line 113. Check, please, this sentence.

“…There are no data on the conservation of mammalian and human species in Bulgaria…”

Perhaps you wanted to say “…There are no data on the infection with HPAI of mammalian and human species in Bulgaria…”

Line 189. A semicolon is omitted between words “…purple other mammals; black…”

Line 206. Correct, please, word in “…interspecific transmission (Figure 3)…”. It must be “…interspecies transmission (Figure 3).”

Line 212. The right name of the reference avian influenza strain is A/Anhui/01/2005 (H5N1) but not H5N1_Human_2005_Anhui1. Correct it please.

Line 260. “…In PB2-M631L [30] it is found in 99% of the sequences in dairy cows, in contrast to birds where it is rarely observed…”

It would be better to write so:

“…Substitution PB2-M631L [29] is found in 99% of the sequences in dairy cows, in contrast to birds where it is rarely observed…”

With best wishes,

Your reviewer

Comments on the Quality of English Language

Unfortunately, I cannot assess the quality of the English language.

Author Response

Dear reviewer,

We accept all your recommendation.

Kind regards.